# Exploring barriers and associated factors in the preparedness for cholera outbreak response in Public Health Centres of Addis Ababa, Ethiopia

Tadios Niguss Derese[1]*, Zekirub Fanta Koyra[2], Lidia Dagne Mario[3], Hiwot Soboksa Mideksa[4], Sisaynesh Angota Dessie[5], Abiye Assefa Berihun[6]

**1** Department of Research and Training, Eka Kotebe General Hospital, Addis Ababa, Ethiopia, **2** Department of Public Health, Gamby Medical and Business College, Addis Ababa, Ethiopia, **3** School of Public Health, Boston University, Boston, Massachusetts, United States of America, **4** Department of Internal Medicine, Blacklion Specialized Hospital, Addis Ababa, Ethiopia, **5** Department of Internal Medicine, Eka Kotebe General Hospital, Addis Ababa, Ethiopia, **6** Johns Hopkins Bloomberg School of Public Health, Baltimore, Maryland, United States of America

\* tadiosnuguss@gmail.com

## Abstract

Cholera remains a significant public health threat in Ethiopia, where recurrent outbreaks disproportionately affect urban populations. Despite the critical role of primary healthcare centers in outbreak response, their level of preparedness (long-term planning, including protocols, training, and resource allocation) and readiness (immediate operational capacity for response) in Addis Ababa is not well understood. An Institutional-based, cross-sectional mixed-methods study was conducted from August to September 2024 across all 101 public health centers in Addis Ababa. Quantitative data were collected using a standardized 14-point WHO checklist (adapted from the Ethiopian national cholera preparedness guideline), which assessed preparedness and readiness and analyzed using binary logistic regression in SPSS version 25. Qualitative data from five key informant interviews and two focus group discussions with health administrators and providers were analyzed thematically to explore barriers.. A total of 101 health centers data found in Addis Ababa were analyzed. Preparedness was measured as meeting ≥75% of indicators (11/14), combining both structural preparedness and operational readiness components. From the total 101 health centers 56 (55.4%) with 95% CI (45.2-65.3) of them were prepared for cholera outbreak response. Having emergency preparedness response plan had significant association with cholera outbreak response preparedness in multivariable analysis at 95% CI ($p < 0.05$). (AOR = 6.88, 95% CI = (1.39-9.89)). The main themes that emerged were financial barriers, lack of cooperation and multi sectorial involvement, and lack of motivation from the health care workers. Cholera outbreak preparedness in health centers of Addis Ababa was found to be 56.7%. Having emergency preparedness response plan in health centers had significant association with cholera outbreak response preparedness. Lack of multi sectorial involvement,

**Data availability statement:** All relevant data are within the paper, and Supporting Information files.

**Funding:** The authors received no specific funding for this work.

**Competing interests:** The authors have declared that no competing interests exist.

financial barrier and lack of motivation from healthcare providers were the main identified barriers towards cholera outbreak response preparedness. The government should develop and implement a comprehensive emergency preparedness response plan specifically addressing cholera outbreaks.

## Introduction

Cholera is an acute diarrheal illness caused by toxigenic strains of *Vibrio cholerae*, typically transmitted through ingestion of contaminated water or food. In severe cases, it can lead to death within hours due to rapid dehydration if not promptly treated [1]. Globally, cholera affects an estimated 1.3 to 4.0 million people each year and is responsible for up to 143,000 deaths, with the overwhelming majority of cases occurring in regions with poor access to water, sanitation, and hygiene (WASH) services [2]. The World Health Organization (WHO) reports that sub-Saharan Africa accounts for more than 99% of global cholera cases, reflecting the persistent infrastructure and health system gaps in the region [3].

Ethiopia remains highly vulnerable to cholera outbreaks. In the past decade, the country has reported recurrent outbreaks in the Somali, Oromia, SNNP, and Amhara regions [4]. Between January and March 2023 alone, Ethiopia recorded 1,002 suspected cases and 28 deaths across 37 woredas, with cases confirmed in four regions [5]. Although Addis Ababa is not a traditional hotspot, its growing population, estimated at over 5 million, and the expansion of informal settlements lacking adequate WASH services have increased the city's risk profile. Several sub-cities have experienced recurrent water supply interruptions, blocked drainage systems, and poor sanitation coverage, all of which heighten the likelihood of cholera transmission [6].

Public health centers (PHCs) serve as the primary access point for health services in Ethiopia and play a central role in outbreak preparedness and response. These centers are responsible for disease surveillance, case management, risk communication, infection control, and coordination with district and regional health authorities. Effective preparedness requires not only infrastructure and supplies but also planning, trained personnel, and strong multisectoral coordination [7]. However, field observations and previous assessments suggest that many PHCs face systemic limitations, including low budget allocation for public health emergencies, inadequate training, and limited staff incentives [8].

A recent WHO assessment found that 86% of African countries had not fully implemented cholera preparedness plans and reported weaknesses in coordination, resource allocation, and early warning systems [9]. Similarly, studies from Uganda, Cameroon, and Kenya found that many health facilities lacked essential outbreak preparedness components, such as updated response plans, pre-positioned supplies, and active collaboration with non-health sectors [10,11].

Although Ethiopia has national guidelines for cholera preparedness and response, the level of implementation at the facility level remains unclear. In Addis Ababa, no recent comprehensive study has assessed the preparedness of public health centers to detect and respond to cholera outbreaks. This gap limits the ability of health planners to prioritize resources or design interventions tailored to actual facility capacities and challenges.

This study was conducted to assess the cholera outbreak preparedness of public health centers in Addis Ababa. It also aimed to identify factors associated with preparedness and explore operational barriers through a mixed-methods approach.

## Methods

### Study area and period

The study was conducted across 101 public health centers under the Addis Ababa City Administration Health Bureau (AACAHB) from August 1 to September 30, 2024. Addis Ababa is the capital city of Ethiopia and the seat of the federal government. The city is home to more than 5 million residents and divided into 11 sub-cities with over 120 woredas. The city operates 101 public health centers, which are responsible for primary healthcare services and are the first line of defense during public health emergencies such as cholera outbreaks.

### Study design

Institution based cross-sectional study supplemented with a Phenomenology qualitative study was conducted.

### Source population

The source population consisted of all public health centers operating under the Addis Ababa City Administration during the study period.

### Study population

The study population included all eligible and functioning public health centers, as well as health professionals engaged in public health emergency preparedness and response.

### Sample size determination

A total population approach was used for the quantitative component, including all 101 health centers. For the qualitative component, five key informants' interviews (KII) and two focus group discussions (FGD) were purposively selected. Qualitative data collection continued until thematic saturation was achieved.

### Sampling techniques

All eligible public health centers were included. For qualitative interviews, purposive sampling was used to select health professionals based on their roles and responsibilities related to cholera response.

### Inclusion criteria

All health centers found in Addis Ababa city administration health bureau.

### Exclusion criteria

Health centers under AACAHB and completely on development construction, currently giving no service, and health centers working as cholera treatment center during the data collection period.

### Data collection tools and techniques

The assessment tool was adapted from both the Ethiopian national cholera preparedness guideline and the WHO cholera readiness checklist, ensuring alignment with long-term preparedness (e.g., planning, training) and immediate readiness (e.g., supplies, case management capacity) [5]. The tool consisted of 14 core criteria:

1.   Availability of a written cholera preparedness and response plan

2.   Presence of trained rapid response teams

3.   Availability of a cholera case definition posted at key service points

4.   Surveillance data analysis for early detection

5.   Functional and timely disease reporting system

6.   Stockpiles of Oral Rehydration Salts (ORS)

7.   Availability of IV fluids (e.g., Ringer's lactate)

8.   Stock of essential antibiotics for cholera treatment

9.   Access to safe water for staff and patients

10.  Functional latrines or toilets for patients and staff

11.  Proper waste disposal system

12.  Availability of handwashing facilities with soap or sanitizer

13.  Availability of personal protective equipment (PPE)

14.  Coordination mechanism with the sub-city health office and other sectors

Each health center was evaluated based on these criteria. Centers were classified as 'prepared' if they met ≥ 75% (11/14) of indicators, a threshold aligned with WHO rapid assessment protocols. To acknowledge the arbitrary nature of binary categorization, sensitivity analyses using 70% and 80% cutoffs was conducted, which showed comparable trends in preparedness levels.

For the qualitative component, data were collected using semi-structured guides. Topics explored included barriers to preparedness, intersectoral coordination, staff capacity, and resource availability. Interviews and discussions were conducted in Amharic, recorded with participant consent, and later transcribed and translated into English.

### Operational definition

Preparedness: Long-term structural and planning aspects (e.g., response plans, trained teams, coordination mechanisms).

Readiness: Immediate operational capacity (e.g., stock availability, case management infrastructure, WASH facilities).

### Data quality control

Data collectors received two days of training. A pre-test was conducted outside the study area. Supervisors reviewed completed forms daily. The principal investigator oversaw the process to ensure consistency and completeness.

### Data analysis

Data were collected by Kobo toolbox and were exported to SPSS v-25 for analysis. Descriptive analysis was presented by tables, graphs and charts. Binary logistic regression technique was used to assess association between independent variables and Cholera outbreak response preparedness. Odds ratio, p-value and 95% confidence interval were used for testing significance and interpretations of result. Variables with p-value of less than 0.05 were considered the cut off point for statistical significance. For qualitative study data were collected through FGD and KII using audiotape record and the

record were transcribed word by word and translated to English language. The thematic approach was followed for displaying the analyses and findings.

### Ethical consideration

Ethical clearance was obtained from the Eka Kotebe General Hospital Institutional Research Ethical Review Committee with a reference number EKGH/IRERC/089/23. Permission was also granted by the Addis Ababa City Administration Health Bureau. Verbal informed consent was obtained from all participants, and each consent event was documented in the presence of a witness. Confidentiality was maintained throughout.

## Results

### Demographic distribution of health centers

A total of 101 health centers data found in Addis Ababa were analyzed. Thirteen (12.9%) health centers were found in Addis Ketema sub city, followed by 11 (10.9%) and 11 (10.9%) in Arada and Gulele sub city. Majority, 57 (56.4%) of health centers had a catchment population of 30, 000–50, 000, followed by 26 (25.7%) and 18 (17.8%) health centers, <30, 000 and > 50, 000 catchment population respectively (Table 1).

### Public health emergency management related characteristics of health centers

All health centers have a public health emergency management (PHEM) officer in their respective health center. Majority 86 (85.1%) of health centers PHEM officer is a health officer, followed by Nurse, MPH and Medical Doctor, 9 (8.9%), 5 (5.0%), and 1 (1.0%) respectively. Majority, 72 (71.3%) of health centers PHEM officer were dedicated only for emergency response. Almost all 95 (94.1%) of PHEM officers took training related to public health emergency management. All health centers have a rapid response team for public health emergency management. Eighty-nine (88.1%) of health centers have an emergency response preparedness plan for cholera outbreak. Majority 88 (87.1%) of health facilities have previous history of having cholera patient in their facility (Table 2).

**Table 1. Demographic distribution of health centers of Addis Ababa, Ethiopia, 2024 (n = 101).**

| Variables | Frequency (%) |
| --- | --- |
| **Health centers in each Sub city** | |
| Addis Ketema | 13 (12.9) |
| Akaki Kality | 9 (8.9) |
| Arada | 11 (10.9) |
| Bole | 6 (5.9) |
| Gulele | 11 (10.9) |
| Kirkos | 9 (8.9) |
| Kolfe keranyo | 7 (6.9) |
| Lideta | 9 (8.9) |
| Nefas Silk | 8 (7.9) |
| Yeka | 10 (9.9) |
| Lemi kura | 8 (7.9) |
| **Catchment population of health centers** | |
| < 30, 000 | 26 (25.7) |
| 30, 000–50,000 | 57 (56.4) |
| >50, 000 | 18 (17.8) |

**Table 2. Public health emergency management related characteristics in public health centers of Addis Ababa, Ethiopia, 2024 (n = 101).**

| Variables | Frequency (%) |
|---|---|
| **PHEM officer in the health center** | |
| Yes | 101 (100) |
| **Profession of PHEM officer** | |
| Nurse | 9 (8.9) |
| HO | 86 (85.1) |
| Medical Doctor | 1 (1.0) |
| MPH | 5 (5.0) |
| **PHEM officer dedicated only for emergency response** | |
| Yes | 72 (71.3) |
| No | 29 (28.7) |
| **PHEM officer took related training** | |
| Yes | 95 (94.1) |
| No | 6 (5.9) |
| **Service year as a PHEM officer** | |
| <= 2 years | 44 (43.6) |
| 3–5 years | 30 (29.7) |
| > 5 years | 27 (26.7) |
| **RR team in the health center** | |
| Yes | 101 (100) |
| **Emergency response preparedness plan** | |
| Yes | 89 (88.1) |
| No | 12 (11.9) |
| **Previous history of cholera faced in health facility** | |
| Yes | 88 (87.1) |
| No | 13 (12.9) |

## Preparedness for cholera outbreak response

The cholera outbreak response preparedness for health facilities were assessed using the cholera kit and calculation tool and the water and sanitation for health facility improvement tool (WASHFIT) assessment tool. From the total 101 health centers 56 (55.4%) with 95% CI (45.2-65.3) of them were prepared for cholera outbreak response (Table 3).

## Factor associated with cholera outbreak response preparedness

In this study emergency preparedness response plan, previous history of cholera faced in the health center, PHEM officer took training were the candidate variables for multivariable analysis.

After adjustment for possible confounders on multivariable analysis having emergency preparedness response plan had significant association with cholera outbreak response preparedness at 95% CI (p < 0.05).

Health centers with emergency preparedness response plan were 6.88 times more likely to have cholera outbreak response preparedness than health centers without emergency preparedness response plan (AOR = 6.88, 95% CI= (1.39-9.89)) (Table 4).

## Challenges and barriers towards cholera outbreak response preparedness

The qualitative component of the study, which included five key informant interviews (KII) and two focus group discussions (FGD), revealed a range of systemic and operational challenges that hinder cholera outbreak preparedness across public

**Table 3. Cholera outbreak response preparedness in public health centres of Addis Ababa, Ethiopia, 2024 (n = 101).**

| Variables | Frequency (%) |
| --- | --- |
| **Presence of isolation unit** | |
| Yes | 95 (94.1) |
| No | 6 (5.9) |
| **Case definition available on poster and wall** | |
| Yes | 90 (89.1) |
| No | 11 (10.9) |
| **Knowledge of suspected case management** | |
| Yes | 100 (99.0) |
| No | 1 (1.0) |
| **Availability of gloves and aprons** | |
| Yes | 97 (96.0) |
| No | 4 (4.0) |
| **Knowledge of suspected case transfer arrangement** | |
| Yes | 98 (97.0) |
| No | 3 (3.0) |
| **Availability of sufficient quantity of ORS** | |
| Yes | 91 (90.1) |
| No | 10 (9.9) |
| **Availability of sufficient quantity of RL** | |
| Yes | 91 (90.1) |
| No | 10 (9.9) |
| **Availability of improved water source** | |
| Yes | 92 (91.1) |
| No | 9 (8.9) |
| **Availability of a functional hand washing station** | |
| Yes | 86 (85.1) |
| No | 15 (14.9) |
| **Availability of sufficient quantity of Chlorine** | |
| Yes | 78 (77.2) |
| No | 23 (22.8) |
| **Availability of well-maintained sterilization equipment** | |
| Yes | 99 (98.0) |
| No | 2 (2.0) |
| **Availability of at least three different bins for waste sorting** | |
| Yes | 86 (85.1) |
| No | 15 (14.9) |
| **Availability of incinerator for infectious waste management** | |
| Yes | 99 (98.0) |
| No | 2 (2.0) |
| **Availability of hand washing station** | |
| Yes | 90 (89.1) |
| No | 11 (10.9) |
| **Overall cholera response preparedness** | |
| Prepared | 56 (55.4) |
| Not prepared | 45 (44.6) |

**Table 4. Multivariable and Bi-variable analysis in the preparedness for cholera outbreak response in public health centers of Addis Ababa, Ethiopia, 2024 (n = 101).**

| Variables | Cholera outbreak response preparedness Frequency (%) | | COR (95% CI) | AOR (95% CI |
|---|---|---|---|---|
| | **Prepared** | **Not prepared** | | |
| **Emergency preparedness response plan** | | | | |
| Yes | 54 (96.4) | 35 (77.8) | 7.71 (1.59–9.32) | 6.88 (1.39–9.89)* |
| No | 2 (3.6) | 10 (22.2) | 1 | 1 |
| **Previous history of cholera faced in the HC** | | | | |
| Yes | 47 (83.9) | 41 (91.1) | 0.50 (0.14–1.77) | 0.66 (0.18–2.37) |
| No | 9 (16.1) | 4 (8.9) | 1 | 1 |
| **PHEM officer took training** | | | | |
| Yes | 54 (96.4) | 41 (91.1) | 2.63 (0.46–7.08) | 1.85 (0.28–7.99) |
| No | 2 (3.6) | 4 (8.9) | 1 | 1 |

P<0.05 =*, P<0.01 =**, P<0.001 =***

health centers in Addis Ababa. The perspectives of participants offered deeper context to the quantitative findings and highlighted entrenched institutional limitations. Three major thematic areas emerged from the qualitative analysis: financial barriers, lack of motivation among health care workers, and inadequate cooperation and multisectoral involvement.

**Theme 1: Financial barrier.** The focused group discussion revealed that the implementation of protocols and guidelines for cholera outbreak response preparedness faced significant challenges. One prominent barrier identified was the lack of adequate financial support for implementation. The informant stated that most protocols and guidelines were not fully implemented due to insufficient funding. This was supported with participants' quotes from the focused group discussion.

"…Most protocols and guidelines were not fully implemented…since it is not supported by adequate finance for implementation" (FGD-1, participant-5).

"…. Lack of budget from the government…. was one of the main challenges that we have faced in cholera outbreak response preparedness…. for instance, our health center did not have any backup budget for this kind of outbreak preparedness. Not only this but also the government needs to set a plan for how to combat emergency situations related to health" (KII-2).

This finding is supported by the quantitative finding mentioned earlier: health centers with emergency preparedness response plans were 6.88 times more likely to have cholera outbreak response preparedness than health centers without emergency preparedness response plans (AOR = 6.88, 95% CI= (1.39-9.89)).

Additionally, the absence of uniformity in implementing these protocols across health centers within each sub-city further compounded the difficulties in adhering to recommended procedures. This was supported with the following quote from participants from a focused group discussion.

"…and also, there is an absence of uniformity in implementing the protocols or guidelines in the same way to all health centers in each sub-city where the difficulties faced in implementing or adhering to recommended protocols or procedures for cholera outbreak response preparedness" (KII-4).

**Theme 2: Lack of motivation from health care workers.** Healthcare professionals expressed a lack of eagerness to participate in cholera outbreak response preparedness. The statement highlighted those previous experiences during

the COVID-19 pandemic, where healthcare professionals did not receive the promised incentives from the government, which have resulted in a diminished motivation to engage in similar efforts for cholera outbreak response preparedness. This sentiment reflects a significant challenge in motivating healthcare workers to actively participate in preparing for and responding to cholera outbreaks. This statement was supported with the following quote from the focused group discussion.

"… staff or healthcare professionals were not eager to participate in cholera outbreak response preparedness due to… healthcare professionals who have participated during the COVID-19 crisis not being incentivized by the government" (FGD-2, participant 7).

"… similarly, for instance, in our health center….no appreciation was given from higher officials or any other immediate boss for health care professionals whenever performing a task that might risk our health… might be our life, for the benefit of the community as a whole" (FGD-2, participant 1).

When healthcare workers lack motivation, it can lead to decreased productivity, lower quality of care, and increased risk of errors or oversights in outbreak response protocols. This can hinder the overall preparedness and response efforts for cholera outbreaks, potentially exacerbating the spread of the disease and its impact on affected communities. This statement is supported with a quote from participants.

"… the government, in collaboration with the Addis Ababa City Administration Health Bureau, needs to give priority to health care providers for the compensation of their efforts to treat and handle patients with compassionate, respectful, and caring behavior… otherwise, the health care providers will lack motivation for their work, and the quality of health will be decreased significantly" (KII-3).

Addressing the lack of motivation among healthcare workers requires a multifaceted approach. It involves understanding the underlying factors contributing to the issue, such as work environment, job satisfaction, and support systems. Implementing strategies to improve motivation, such as providing adequate training, recognition, and opportunities for professional growth, can help boost healthcare workers' morale and engagement.

**Theme 3: Lack of cooperation and multi-sectoral involvement.** During the focused group discussion and key informant interview conducted to assess the challenges of cholera outbreak response preparedness in health centers, participants highlighted a concerning lack of multi-sectoral involvement. Specifically, they expressed that there was a notable absence of cooperation from other healthcare workers, resulting in an overreliance on the Public Health Emergency Management (PHEM) officer. This was supported with the following quotes from the participant in the key informant interview.

"… most sectors ignored the health-related conditions for only the health sector; for instance, when a pandemic happens like COVID-19 or any other case in Ethiopia, it is unlikely to get support from other sectors…. Budget support or any other material support is crucial for the health sector from other sectors to combat the pandemic that happens" (KII-1).

Participants reported that other health professionals often considered the PHEM officer as the sole individual responsible for managing all aspects of cholera outbreak response preparedness. This was evident in their statements, such as "Your case is coming" when a cholera case arrived at the health facility, indicating that the PHEM officer was expected to handle the situation alone.

"…as a PHEM officer….one of the funniest experiences I have encountered, mostly from other health professionals in our health center, is that nurses or midwives…said to me that your case is coming whenever a cholera case arrived at our health facility" (FGD-1, Participant 3).

Furthermore, participants emphasized that the perception of the PHEM officer as the only dedicated person for managing cholera outbreaks led to an expectation of their availability and commitment to work 24 hours during emergency responses. This reliance on a single individual not only placed an excessive burden on the PHEM officer but also limited the involvement of other healthcare professionals who could contribute to an effective multi-sectoral response.

## Discussion

The study assessed both preparedness and readiness in 101 public health health centers, though the primary focus was on structural preparedness (plans, training). However, the readiness components (supplies, case management) were also critical in determining overall outbreak response capacity. Of these, 56 centers (55.4%) with 95% CI: (45.2–65.3) were classified as prepared based on a standardized threshold. This indicates that nearly half of the city's health centers are not adequately equipped to respond effectively to cholera outbreaks, raising important concerns for urban health system resilience.

A key factor significantly associated with preparedness was the presence of a written emergency preparedness and response plan. Health centers with such plans were substantially more likely to meet the preparedness threshold. This finding supports guidance from the World Health Organization, which emphasizes that clearly defined emergency plans are central to effective outbreak control [3]. Ethiopia's national guideline for cholera management also highlights the importance of facility-level planning to enable early detection, coordination, and response [5].

In comparison to other countries in the region, Addis Ababa demonstrated a relatively higher level of preparedness. For instance, a study conducted in Namayingo District, Uganda, reported that only 38.1 percent of facilities were adequately prepared [12]. Similarly, in Cameroon, preparedness was 50% in the Far North and just 20% in the Littoral region [13]. These differences may reflect the advantages of urban settings, such as better infrastructure, closer proximity to health authorities, and improved access to resources. However, the preparedness of only slightly more than half of the centers also reflects systemic challenges that must be addressed.

Qualitative findings revealed that these challenges were largely driven by three primary barriers to cholera preparedness: financial constraints, limited multisectoral collaboration, and low staff motivation. Financial limitations were frequently mentioned. Most health centers lacked dedicated funding for outbreak preparedness, which constrained their ability to procure supplies, deliver training, or conduct simulations. This aligns with broader regional evidence that inconsistent or insufficient funding remains one of the primary challenges to strengthening outbreak preparedness in Africa [9].

Multisectoral collaboration was also lacking. Respondents described a situation where responsibility for outbreak preparedness often fell solely on Public Health Emergency Management (PHEM) officers, while other departments remained uninvolved. Such fragmented responses are inconsistent with both national and international recommendations, which call for coordinated engagement among sectors including water and sanitation, education, and local administration [3,14].

Low healthcare worker motivation was a third critical barrier. Participants reported that staff morale had declined, particularly after unfulfilled commitments during the COVID-19 pandemic. Many felt their contributions were undervalued, which negatively affected their willingness to participate in future outbreak preparedness activities. Other studies have shown that lack of recognition and reward can reduce health workers' engagement in emergency response planning and training [15].

In addition to morale, low motivation may contribute to poor compliance with infection prevention and control practices. Disengaged staff are less likely to adhere to protocols such as proper hand hygiene, consistent use of personal protective equipment, and patient isolation procedures. These lapses increase the risk of healthcare-associated transmission, a problem observed during previous cholera outbreaks [14,16].

As a limitation the study highlighted these points, the binary preparedness classification (using a 75% threshold), while operationally useful, may obscure granular differences between facilities scoring near the cutoff. Checklist items were equally weighted despite varying clinical urgency (e.g., ORS vs. incinerator access); future studies could apply

expert-derived weights. Findings reflect health worker and administrator perspectives; future research should incorporate community voices to triangulate barriers.

While this study assessed facility-level cholera preparedness, it did not evaluate critical community-based components of outbreak readiness, including early warning systems, risk communication strategies, or household prevention practices. This omission, though consistent with our institutional assessment focus, means our findings cannot speak to the community engagement aspects emphasized in WHO's holistic preparedness framework. Future research should integrate both facility and community indicators for a complete evaluation of outbreak resilience.

## Conclusion

The study findings reveal that just over half (55.4%) of health centers in Addis Ababa met the studies preparedness threshold, suggesting room for improvement despite some foundational capacity. The presence of an emergency preparedness response plan in these health centers was significantly associated with improved preparedness for cholera outbreak response. However, certain barriers and challenges were identified, including a lack of cooperation and multi-sectorial involvement, financial barriers, and a lack of motivation among healthcare providers. To enhance cholera outbreak response preparedness in health centers, it is crucial to address these challenges. Efforts should be focused on promoting collaboration and multi-sectorial involvement, addressing financial barriers, and addressing motivational factors among healthcare providers. By overcoming these barriers, the overall preparedness and response to cholera outbreaks can be further improved in health centers in Addis Ababa.

To strengthen cholera preparedness, health authorities should ensure that all facilities have actionable response plans, trained personnel, and the capacity to regularly monitor surveillance data. Emphasis should also be placed on dedicated cholera preparedness budgets at the sub-city, tiered financing model where high-risk facilities receive priority funding, Integration with WASH programs to leverage existing water/sanitation budgets.

To enhance multisectorial coordination, quarterly 'outbreak readiness committees' meetings, Joint simulations (tabletop exercises with utility companies to test water contamination responses), clear accountability frameworks (Addis Ababa's Health Bureau tracking WASH compliance in facilities) should be strengthened.

Future studies should consider evaluating the long-term impact of these interventions and explore strategies for integrating preparedness across broader health system functions, and with larger samples of unprepared facilities to improve precision.

## Supporting information

**S1 Checklist. PLOS checklist on inclusivity in global research.**
(DOCX)

**S1 Data. SPSS raw data for the qualitative component of the study.**
(SAV)

## Acknowledgments

The authors acknowledge the data collectors, study participants, and the management bodies of the hospitals.

## Author contributions

**Conceptualization:** Tadios Niguss Derese, Zekirub Fanta Koyra, Hiwot Soboksa Mideksa.

**Data curation:** Tadios Niguss Derese, Lidia Dagne Mario, Sisaynesh Angota Dessie, Abiye Assefa Berihun.

**Formal analysis:** Tadios Niguss Derese, Zekirub Fanta Koyra, Abiye Assefa Berihun.

**Investigation:** Tadios Niguss Derese, Zekirub Fanta Koyra.

**Methodology:** Tadios Niguss Derese, Zekirub Fanta Koyra, Abiye Assefa Berihun.

**Project administration:** Tadios Niguss Derese.

**Resources:** Tadios Niguss Derese, Lidia Dagne Mario, Sisaynesh Angota Dessie.

**Software:** Tadios Niguss Derese, Zekirub Fanta Koyra, Hiwot Soboksa Mideksa.

**Supervision:** Tadios Niguss Derese.

**Validation:** Tadios Niguss Derese, Zekirub Fanta Koyra, Hiwot Soboksa Mideksa.

**Visualization:** Tadios Niguss Derese, Lidia Dagne Mario, Abiye Assefa Berihun.

**Writing – original draft:** Tadios Niguss Derese, Zekirub Fanta Koyra.

**Writing – review & editing:** Tadios Niguss Derese, Zekirub Fanta Koyra, Lidia Dagne Mario, Hiwot Soboksa Mideksa, Sisaynesh Angota Dessie, Abiye Assefa Berihun.

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
