## [Editor Report · Decision Letter 0]

29 Jul 2025

PGPH-D-25-02066

Exploring Barriers and Associated Factors in the Preparedness for Cholera Outbreak Response in Public Health Centres of Addis Ababa, Ethiopia

Dear Dr., DereseTadios Niguss

Thank you for submitting your manuscript to PLOS Global Public Health. After careful consideration, we feel that it has merit but does not fully meet PLOS Global Public Health’s publication criteria as it currently stands. Therefore, we invite you to submit a revised version of the manuscript that addresses the points raised during the review process.

We look forward to receiving your revised manuscript.

Kind regards,

Laston Gonah

Academic Editor

Journal Requirements:

Additional Editor Comments (if provided):

I have decided to hold off on inviting reviewers at this time, as I believe the paper requires significant revisions to enhance its clarity and quality. Here are the reasons:

1. Conceptual Confusion Between "Preparedness" and "Readiness". The paper consistently uses “preparedness” without defining it or distinguishing it from “readiness.” In public health, “preparedness” generally refers to long-term planning (e.g. training, protocols, resource allocation), while “readiness” refers to immediate operational ability to respond. By conflating or ignoring the distinction, the study may overestimate the system’s actual capability during an outbreak. The mention of adapting a checklist from the Ethiopian guideline for cholera “preparedness to assess readiness” creates ambiguity about the tool’s origin (Ethiopian government or WHO because sections mention the 14—point WHO checklist was used), and purpose, potentially conflating preparedness and response concepts. It remains unclear which questions assess preparedness, and which ones assess readiness, or at last operational definitions of the terms.

2. Over-Reliance on Binary Categorization

- Health centers were labelled as either “prepared” or “not prepared” based on meeting 75% of checklist items. This threshold is arbitrary and lacks validation or justification in the paper. After all, a health center with 10/14 items may be nearly as capable as one with 11/14 but is considered unprepared - why did the authors choose binary categorization over all other options?

- It seems the 14 items were presumed to have equal weights in the assessment of preparedness, since no weighting of items was mentioned. Instance, can ORS availability and incinerator access be equal in outbreak response relevance?

3. Weakness in Statistical Analysis

- Only one variable (presence of preparedness plan) was found statistically significant in multivariable analysis. Despite collecting many variables (e.g., training, previous cholera experience), the model explains little variance. This suggests either limited variability in preparedness scores, or inadequate power to detect meaningful associations.

- Confidence intervals are wide, suggesting unstable estimates, for example, AOR of 6.88 (1.39–9.89) indicates lack of precision.

4. Methodological Concerns in Qualitative Component.

- Small sample size: Only 5 key informant interviews (KII) and 2 FGDs across 101 facilities, and no mention of the principle of saturation is included.

- Limited triangulation: No patient or community perspective is included, or at least some acknowledgement of this limitation.

5. All the limitations that come with the study must be acknowledged, for example, consideration of community engagement. Public health preparedness includes community awareness, early warning systems, and risk communication. These are entirely omitted, though they are part of WHO’s preparedness indicators.

6. Overstatement in Conclusion. The conclusion claims "preparedness is relatively high," but only 55.4% of facilities met the criteria. This is not a majority worth characterizing as relatively high, especially for an urban capital with outbreak history.

7. Lack of Actionable Policy Recommendations. While the paper identifies barriers (financial, motivation, multisectoral coordination), no concrete solutions are proposed. Example: No costed plan, incentive strategies, or examples of successful multisectoral collaboration.
---

## [Decision Letter · Decision Letter 1]

20 Aug 2025

Exploring Barriers and Associated Factors in the Preparedness for Cholera Outbreak Response in Public Health Centres of Addis Ababa, Ethiopia

PGPH-D-25-02066R1

Dear Prof/Dr Tadios Niguss Derese and team,

We are pleased to inform you that your manuscript 'Exploring Barriers and Associated Factors in the Preparedness for Cholera Outbreak Response in Public Health Centres of Addis Ababa, Ethiopia' has been provisionally accepted for publication in PLOS Global Public Health.

Best regards,

Laston Gonah

Academic Editor

Reviewer Comments (if any, and for reference):

Reviewer's Responses to Questions

**Comments to the Author**

1. If the authors have adequately addressed your comments raised in a previous round of review and you feel that this manuscript is now acceptable for publication, you may indicate that here to bypass the “Comments to the Author” section, enter your conflict of interest statement in the “Confidential to Editor” section, and submit your "Accept" recommendation.

Reviewer #1: All comments have been addressed

2. Does this manuscript meet PLOS Global Public Health’s publication criteria? Is the manuscript technically sound, and do the data support the conclusions? The manuscript must describe methodologically and ethically rigorous research with conclusions that are appropriately drawn based on the data presented.

Reviewer #1: Yes

3. Has the statistical analysis been performed appropriately and rigorously?

Reviewer #1: Yes

4. Have the authors made all data underlying the findings in their manuscript fully available (please refer to the Data Availability Statement at the start of the manuscript PDF file)?

Reviewer #1: Yes

5. Is the manuscript presented in an intelligible fashion and written in standard English?

Reviewer #1: Yes

6. Review Comments to the Author

Reviewer #1: Thank you for writing this insightful journal article that provides essential context about cholera preparedness in Ethiopia from 101 health centres across the country. Particularly valuable to contextualise your findings was the qualitative insights, which showed, inter alia, that the lack of a specific budget impedes their responses, and that a lack of appreciation for efforts may have caused low motivation. I think this is an excellent article and I will be making a recommendation for publication.

7. PLOS authors have the option to publish the peer review history of their article (what does this mean?). If published, this will include your full peer review and any attached files.

**Do you want your identity to be public for this peer review?** For information about this choice, including consent withdrawal, please see our Privacy Policy.

Reviewer #1: **Yes: **Fifa A Rahman
